# Relative Strength, but Not Absolute Muscle Strength, Is Higher in Exercising Compared to Non-Exercising Older Women

**DOI:** 10.3390/sports7010019

**Published:** 2019-01-10

**Authors:** Claudio de Lira, Valentine Vargas, Wallace Silva, André Bachi, Rodrigo Vancini, Marilia Andrade

**Affiliations:** 1Laboratório de Avaliação do Movimento Humano, Faculdade de Educação Física e Dança, Universidade Federal de Goiás, Goiânia 74690-900, Brazil; 2Departamento de Fisiologia, Escola Paulista de Medicina, Universidade Federal de São Paulo, São Paulo 04023-062, Brazil; valentine.vargas@unifesp.br (V.V.); wallace-89@hotmail.com (W.S.); marilia.andrade@unifesp.br (M.A.); 3Departamento de Otorrinolaringologia e Cirurgia de Cabeça e Pescoço, Escola Paulista de Medicina, Universidade Federal de São Paulo, São Paulo 04039-002, Brazil; allbachi77@gmail.com; 4Centro Universtario Evangelica, Anápolis 75083-515, Brazil; 5Centro de Educação Física e Desportos, Universidade Federal do Espírito Santo, Vitória 29060-220, Brazil; rodrigo.vancini@ufes.br

**Keywords:** aging, isokinetic dynamometer, knee, body composition, exercise

## Abstract

Exercise has been suggested for older adults. However, there is no consensus whether exercising older adults present better strength levels and body composition indexes compared with inactive counterparts. Our aim was to compare absolute and relative isokinetic muscular knee strength and body composition between exercising and non-exercising older women. Exercising (n = 20) and non-exercising (n = 21) groups were evaluated for body mass index (BMI), body composition, and isokinetic muscular knee strength. BMI (*p* = 0.005), total body mass (*p* = 0.01), fat mass (*p* = 0.01), and fat mass percentage (*p* = 0.01) were higher in non-exercising women, and the lean mass percentage was lower in the non-exercising group (*p* = 0.01). Isokinetic extensor and flexor knee muscle strength for dominant limbs presented higher peak torque values when corrected for total body mass (Nm·kg^−1^) in the exercising group (*p* < 0.05). Exercising older women presented better body composition and higher strength relative to total body mass, but not maximum absolute strength.

## 1. Introduction

Increasing life expectancy elevates the incidence and prevalence of chronic diseases associated with aging, such as diabetes mellitus, osteoporosis, cancer, and obesity [1]. Obesity may contribute substantially to disability and chronic diseases by impairing an individual’s ability to perform independently the activities of daily living, and thus compromise quality of life [2]. For good quality of life, it is essential that older adults have a good body composition (e.g., low fat mass percentage) and muscular strength [2].

However, the natural aging process causes several physiological changes, including structural and functional changes in the musculoskeletal system. Previous studies demonstrated a loss of 1–2% lean mass and 1.5–5.0% strength per year in individuals older than 50 years. This muscle mass loss is a multifactorial process characterized by a reduced number of muscle fibers as a consequence of denervation and reinnervation processes associated with aging [3,4,5]. This combination of low muscle mass and low muscle strength has been termed sarcopenia [6]. 

More recently, osteosarcopenic obesity syndrome (OSO) has been identified as a condition existing alongside sarcopenia, osteopenia/osteoporosis, and obesity [7]. It is important to note that obesity is only diagnosed if the subject presents a body mass index higher than 30 kg/m^2^. However, the distribution of fat mass is also important as fat infiltration on muscle mass or bone is also a harmful effect of aging [7]. 

It has been suggested that the loss of skeletal muscle strength is even more related to functional decline than the loss of muscle mass [8]. In this direction, in a recent review, Shaw et al. [9] argued for the urgent need for a consensual definition of sarcopenia. The European Working Group on Sarcopenia in Older People (EWGSOP) [10] utilized grip strength (<30 kg in men and <20 kg in women) as a criteria to diagnose low muscular strength. Although this criterion has been widely used, it refers to absolute strength and, considering that individuals may have quite different body dimensions and compositions, a strength evaluation that takes into account the strength relative to total body mass will be of interest. Considering the fat infiltration on muscle mass, resulting from obesity and aging [11,12], it is possible that the peak torque generated per kilogram of muscle mass is modified if the mass muscle is infiltrated by fat. Therefore, information about the muscular strength relative to lean mass may also provide important knowledge about the ability of lean tissue to develop tension. As for maximal oxygen uptake, which is also expressed relative to total body mass when comparing exercise tolerance [13], muscular strength can also be expressed relative to total body mass.

Maximal absolute muscle strength refers to the maximum tension level that a muscle group can produce. This maximum tension will be sufficient to generate a motion, depending on the body mass that needs to be mobilized. In this context, a woman may present a high level of absolute muscle strength, but, depending on whether she presents a high total body mass, her strength level may be insufficient to independently perform the activities of daily living. Thus, another interesting way to assess an individual’s ability to perform a task is to divide muscular peak torque by the total body mass [14]. Another useful strength measure is the peak torque relative to lean mass. 

Recognizing the importance of maintaining the integrity of the musculoskeletal system, exercise has been suggested for people diagnosed with OSO [15] to maintain or improve body mineral density and muscle strength, to prevent falls, and to reduce total body mass and inflammation [16,17,18]. Indeed, a regular exercise program would have positive effect on sarcopenic muscle through improving muscle mass and strength [19]. However, along with absolute strength levels, relative strength to total body mass may also provide important information about muscular function and this issue has not been reported in previous studies. In this sense, it is very important to know whether an exercising program performed for more than six months affects absolute strength, strength relative to total body mass, and relative to lean mass. 

Thus, the aim of this study was to compare absolute strength, relative to total body mass and relative to lean mass muscular strength—of the knee flexor and extensor muscles—and body composition (i.e., total, lean, and fat body mass), of exercising (more than 6 months) and non-exercising older women.

We hypothesized that women who are chronically involved in exercising activities would present a lower fat mass and a higher fat-free mass percentage, and thereby the relative to total body mass percentage should be higher in active women.

## 2. Materials and Methods

### 2.1. Sample

Forty-one women, 60–80 years old, were invited to participate in the study. Twenty-one women, who were not involved in any exercise program, comprised the non-exercising group (NEG). To be classified as non-exercising, the participants should not practice any exercise program for more than 30 min each day. The participants were selected from the Federal University of São Paulo (UNIFESP, São Paulo, Brazil) community. This study utilized non-random, consecutive participant enrollment and group assignment. The first 21 participants who looked for the study were selected to participate. Twenty women were participants in the regular exercise program for older people developed at Clube Escola Ibirapuera of São Paulo, and they composed the exercising group (EG). 

Inclusion criteria were as follows: An age between 60–80 years; and medical authorization to perform maximum physical effort. Exclusion criteria were as follows: The use of corticosteroids and/or anabolic hormones; being human immunodeficiency virus (HIV) positive; or suffering from diabetes mellitus, cancer, chronic infections, neurological diseases, orthopedic diseases in the lower limbs, or cardiovascular or metabolic diseases which would contraindicate the realization of exercise activities. 

### 2.2. Experimental Design

We conducted a cross-sectional, observational study featuring a between-subjects design. The study was developed in three stages: (1) A pre-participation clinical examination to verify whether the volunteers met the inclusion criteria; (2) Anthropometric measurements and a body composition evaluation; and (3) Knee flexor and extensor muscle isokinetic strength tests. All stages of the study were performed in the mornings. As it was a cross-sectional study, subjects were evaluated only once. Therefore, the EG was not evaluated prior the exercise program.

### 2.3. Anthropometric Assessment

Body composition and body mass index (BMI) were analyzed. A BMI ≥ 25 kg/m^2^ indicated overweight, while a BMI ≥ 30 kg/m^2^ indicated obesity [20]. 

Body composition was assessed by dual-energy X-ray absorptiometry (DXA, software version 12.3, Lunar DPX, Madison, WI, USA). DXA is based on the concept of double beams of radiation (X-rays) with the ability to measure bone mineral content, and estimate fat mass and total fat-free mass. The same professional performed all of the measurements and calibrations. DXA measurements have been validated against corresponding ones from computed tomography [21] and magnetic resonance imaging [22].

### 2.4. Muscular Isokinetic Assessment

The muscular strength test was preceded by a warm-up exercise—5-min on a cycle ergometer at 25 W and 50–60 rotations per minute and light stretching exercises for lower limbs [23]. The strength assessment was performed for both lower limbs on an isokinetic dynamometer (Biodex Medical Systems Inc., Shirley, NY, USA). The dominant lower limb was determined as the limb with which the participants preferred to kick a ball. A seated position was adopted for the isokinetic strength test. Hips were flexed at 85°. Trunk, waist and femur were fixed with standard stabilization strapping to avoid additional different movements of the knee flexion and extension to be tested. 

The axis of the dynamometer and the lateral femoral condyle were visually aligned. Gravity correction was performed according to the manufacturer’s specifications. Full knee extension was considered as 0°, and the knee range of motion tested was from 5 to 95° [30]. 

The strength test consisted of 5 maximum repetitions at 1.05 and 3.14 rad·s^−1^ (in this order), and the results were stored for analysis. These repetition numbers have demonstrated high reliability for isokinetic testing [24]. Prior to the test, participants were instructed to perform five submaximal repetitions with each test velocity for familiarization. Between the test velocities a 60 s rest was given and between lower limbs a 3 min rest. Consistent verbal commands were given by the same trained experienced examiner during the entire strength test. The parameters evaluated were absolute peak torque (Nm), peak torque of the hamstring and quadriceps muscles relative to total body mass (Nm·kg^−1^), and peak torque relative to fat-free mass (Nm·kgFFM^−1^). Peak torque relative to total body mass and relative to fat-free mass were calculated by dividing peak torque (Nm) values by total body mass (Kg) and by fat-free mass (KgFFM), respectively. 

### 2.5. Training Program

The EG was composed of women who were physically active for at least 6 months (and less than 5 years) prior to the study’s commencement. The volunteers held training sessions 3 times a week, 1 h per session. The program consisted of aerobic and strength exercises. Aerobic exercises were developed at 60–70% of maximal heart rate as estimated by the equation 220 – age. Each aerobic training session lasted 30 min, and the sessions consisted of dance, step, and jumping exercises. Aerobic training was followed by strength exercises, which involved 3 exercises for lower limbs, 3 for upper limbs, and 3 for the trunk. Exercises were performed in 2 sets of 10 to 20 repetitions (lasting a maximum of 30 min) using washers, rubber bands, or free weights. The aerobic training intensity was monitored using a heart rate monitor. An experienced coach supervised each exercise training session (Polar FT1 model, Polar, Finland).

### 2.6. Ethical Aspects

All participants were informed of the potential risks and benefits of the study and signed an informed consent form to take part. All experimental procedures were approved (number 213/2014) by the Cruzeiro do Sul University Human Research Ethics Committee and conformed to the principles outlined in the Declaration of Helsinki.

### 2.7. Statistical Analyses

Results are presented as mean ± standard deviation (SD). All variables were normally distributed according to the Shapiro-Wilk test, and they also exhibited homogeneous variability as determined by the Levene test. Comparison of isokinetic and body composition data between groups was performed using an unpaired Student’s *t* test. In order to verify the Type II error magnitude (probability to falsely infer the absence of something that is present) for each variable evaluated, power analysis was conducted after data collection. All statistical analysis was carried out for two-tailed significance at the 5% level using Statistica software (Statsoft, Inc., Tulsa, OK, USA, version 6.0 for Windows). 

## 3. Results

There were no significant differences in age, height or lean mass between groups (Table 1). However, fat mass, total body mass, fat mass percentage, and BMI were higher in the NEG and the lean mass percentage was lower in the NEG compared to the EG (Table 1). According to BMI categories, in the NEG, five participants were classified as normal, 12 as overweight, and four as obese, and in the EG there were 13 participants classified as normal, six as overweight, and one as obese. 

There was no significant difference between groups with regards to absolute peak torque values for either flexor or extensor muscles for both angular speeds used (Table 2).

Muscular isokinetic testing showed higher strength values relative to body mass for almost all the variables measured in the EG compared to the NEG. Extensor and flexor muscles in the dominant limb assessed at both angular speeds were higher in the EG. The only similarity between groups was for extensor peak torque values at 1.05 rad·s^−1^ and flexor peak torque values at 3.14 rad·s^−1^ for non-dominant limbs (Table 3). 

Peak torque values relative to fat-free mass were significantly lower for non-dominant knee extensor muscles assessed at 1.05 and 3.14 rad·s^−1^ in the NEG compared to the EG (Table 4).

## 4. Discussion

The aim of this study was to compare absolute strength, relative to total body mass and relative to lean mass muscular strength and body composition, in exercising and non-exercising older women. Although peak torque values were not different between groups, the EG had higher peak torque values relative to total body mass and relative to lean mass (for non-dominant extensor muscles), showing a better physical fitness. Additionally, we found that exercising women had a higher percentage lean body mass. The EG showed significantly lower total body mass and fat mass than the NEG, but the groups were not different with regards to total lean body mass (kg). Further, since the EG total body mass was significantly lower than the NEG, with the same absolute lean body mass (kg), the EG had a higher lean mass percentage and, therefore, a better body composition. As it was a cross-sectional study, it was not possible to affirm whether the exercise program improved body composition or muscular relative strength. Instead, the discussion was based on inter-group differences. 

Aside from the adverse effect of lean mass loss, an increase in fat mass is associated with a significant reduction of physical capacity in older adults [25,26]. Increased body fat mass appears to be associated with fat and connective tissue infiltration in muscle in response to muscle fiber atrophy and consequent loss of muscle performance [27,28]. Additionally, a higher fat mass implies the need for greater absolute strength levels to perform the same physical activity, and participants with a higher fat mass will expend more energy to perform the same task [29]. 

In a previous study, Zamboni et al. [30] showed that even when holding physical activity and body mass levels constant as age increased, fat mass was elevated and lean body mass was decreased. Our results revealed that the EG presented a better body composition, a phenomenon which helps maintain older adults’ functional capacity, and thus quality of life. Previous studies demonstrated that strengthening programs generate strength gains in young individuals and in older adults, but in older persons gains are of a smaller magnitude [31]. In the present study, both groups exhibited the same muscle mass (kg) and the same absolute peak torque (Nm) in thigh muscles—findings which show the positive effects of exercise programs since lower strength values would be expected for lighter subjects. 

In order to compare individuals with different body mass, peak torque relative to total body mass (Nm·kg^−1^) should be used since it better compares physical fitness capacity. In this context, the peak torque relative to body mass values for knee extensor and flexor muscles were higher for the EG. As a result, we concluded that the EG had better muscular strength and functional capacity than the NEG. Similarly, Araujo et al. [26] found that older persons with higher BMIs and lower physical activity levels presented lower functional capacity and were more vulnerable to lower quality of life levels. 

According to BMI, the NEG participants were significantly heavier than the EG subjects. Indeed, the NEG participants were classified as overweight, while the EG participants were classified as eutrophic. Previous data indicate that fat infiltrates skeletal muscle in obese people, and this infiltration reduces the contractile component of the total muscle volume and damages strength development [32,33,34,35]. In order to verify whether there was any difference between groups with regards to muscle mass capacity to generate torque, peak torque values relative to fat-free mass (Nm·Kg FFM^−1^) were compared between groups. The results showed that these values for non-dominant extensor muscles, assessed at both test speeds, were significantly higher in the EG compared to the NEG. This data corroborates the hypothesis that fat mass affects muscle mass activation, and also confirms the previous finding that exercise programs maintain the muscular capacity to generate strength [36]. On the other hand, extensor and flexor muscle strength (Nm·Kg FFM^−1^) for the dominant limb were no different between groups. However, the power analysis was lower than 80%. Therefore, it is possible that a larger sample size would reveal differences between groups. We suggest a clinical trial to better understand the effects of exercise progras for older women. Another possible reason for the lack of difference between groups in relation of peak torque relative to lean mass is that the fat mass infiltration in skeletal muscle (myosteatosis) increases with aging even if subjects are not obese [37]. Therefore, both groups of the present study should present some level of skeletal muscle fat infiltration. In order to clarify this point, future studies that evaluate body mass composition through computed tomography imaging, which is considered the gold standard method to evaluate the fat mass infiltrate in skeletal muscle [38], can be performed. 

### Study Limitation and Strengths

This study was not a clinical trial, and subjects were already participating in exercising programs prior to the study’s initiation. Thus, it was not possible to control for the exercise programs that had been performed previously, and we do not have precise information about subjects’ exercise progression. Muscular performance was evaluated with the isokinetic dynamometer, which does not replicate the muscle torques used in functional activities. Nevertheless, it is considered a reliable and valid instrument for muscle force testing [39,40,41,42]. Moreover, we were unable to objectively assess muscle tissue composition. However, we were able to obtain valid measures of body composition with DXA, which is also considered a reliable and valid instrument for measuring body composition [21].

## 5. Conclusions

Older exercising women had better body composition and relative muscle strength compared to older non-exercising women. These differences probably enable greater functional capacity for exercising women. Considering the increasing life expectancy of the world population and the increasing incidence of obesity, the findings of this study have an important social contribution. 

## Figures and Tables

**Table 1 sports-07-00019-t001:** Sample characteristics of exercising (EG) and non-exercising groups (NEG).

Variables	EG (n = 20)	NEG (n = 21)	*p* Value	Power Analyses
Age (years)	70.9 ± 5.1	69.6 ± 4.8	0.39	0.71
Height (cm)	155.3 ± 6.6	156.1 ± 6.4	0.70	0.81
Lean mass (kg)	33.7 ± 3.4	35.1 ± 3.7	0.19	0.64
Fat mass (kg)	23.5 ± 8.1	30.1 ± 7.9	0.01 *	0.58
Total body mass (kg)	59.3 ± 10.6	67.5 ± 10.0	0.01 *	0.54
Total body lean mass (%)	57.8 ± 7.1	52.6 ± 5.6	0.01 *	0.57
Total body fat mass (%)	40.1 ± 7.5	45.6 ± 5.8	0.01 *	0.58
BMI (kg·m^−2^)	24.4 ± 3.1	27.7 ± 3.9	0.005 *	0.61

Data expressed as mean ± standard deviation (SD); * *p* < 0.05; BMI: body mass index; power analyses refer to Type II error.

**Table 2 sports-07-00019-t002:** Absolute isokinetic peak torque (Nm) for flexor and extensor knee muscles in exercising (EG) and non-exercising groups (NEG).

Variables	EG (n = 20)	NEG (n = 21)	*p* Value	Power Analyses
1.05 rad·s^−1^				
Dominant extensor muscles	75.1 ± 9.9	73.2 ± 23.9	0.75	0.84
Non-dominant extensor muscles	78.1 ± 10.1	70.7 ± 26.5	0.25	0.70
Dominant flexor muscles	34.8 ± 7.3	31.5 ± 13.3	0.34	0.72
Non-dominant flexor muscles	36.1 ± 7.8	33.0 ± 12.7	0.35	0.70
3.14 rad·s^−1^				
Dominant extensor muscles	51.0 ± 9.3	48.3 ± 14.5	0.49	0.75
Non-dominant extensor muscles	52.1 ± 7.9	45.7 ± 13.5	0.07	0.63
Dominant flexor muscles	26.9 ± 6.7	24.5 ± 9.8	0.37	0.72
Non-dominant flexor muscles	26.7 ± 5.9	25.4 ± 10.0	0.60	0.77

Data expressed as mean ± SD. Power analyses refer to Type II error.

**Table 3 sports-07-00019-t003:** Relative isokinetic peak torque (Nm·kg^−1^) for flexor and extensor knee muscles in exercising (EG) and non-exercising groups (NEG).

Variables	EG (n = 20)	NEG (n = 21)	*p* Value	Power Analyses
1.05 rad·s^−1^				
Dominant extensor muscles	127.9 ± 22.9	110.5 ± 31.3 *	0.04 *	0.59
Non-dominant extensor muscles	124.7 ± 31.2	107.3 ± 35.3	0.10	0.64
Dominant flexor muscles	59.6 ± 15.3	47.4 ± 17.0 *	0.02 *	0.61
Non-dominant flexor muscles	61.6 ± 14.8	47.9 ± 19.2 *	0.01 *	0.56
3.14 rad·s^−1^				
Dominant extensor muscles	87.8 ± 23.8	73.4 ± 19.4 *	0.04 *	0.62
Non-dominant extensor muscles	88.8 ± 18.2	68.6 ± 18.5 *	>0.01 *	0.58
Dominant flexor muscles	46.5 ± 15.2	36.8 ± 12.9 *	0.03 *	0.61
Non-dominant flexor muscles	45.6 ± 11.9	38.1 ± 12.8	0.06	0.64

Data expressed as mean ± SD; * *p* < 0.05; power analyses refer to Type II error.

**Table 4 sports-07-00019-t004:** Peak torque relative to fat-free mass (Nm·Kg FFM^−1^) for flexor and extensor knee muscles in exercising (EG) and non-exercising groups (NEG).

Variables	EG (n = 20)	NEG (n = 21)	*p* Value	Power Analyses
1.05 rad·s^−1^				
Dominant extensor muscles	2.24 ± 0.38	2.07 ± 0.56	0.24	0.66
Non-dominant extensor muscles	2.32 ± 0.29	1.99 ± 0.65	0.04 *	0.61
Dominant flexor muscles	1.04 ± 0.25	0.89 ± 0.33	0.10	0.63
Non-dominant flexor muscles	1.07 ± 0.22	0.93 ± 0.33	0.12	0.66
3.14 rad·s^−1^				
Dominant extensor muscles	1.53 ± 0.36	1.36 ± 0.34	0.13	0.65
Non-dominant extensor muscles	1.56 ± 0.30	1.29 ± 0.32	0.01 *	0.64
Dominant flexor muscles	0.81 ± 0.23	0.69 ± 0.23	0.11	0.66
Non-dominant flexor muscles	0.79 ± 0.18	0.71 ± 0.25	0.26	0.70

Data expressed as mean ± SD; * *p* < 0.05; power analyses refer to Type II error.

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
