# Peer review of "Relative Strength, but Not Absolute Muscle Strength, Is Higher in Exercising Compared to Non-Exercising Older Women"

_sports, 2019, doi:10.3390/sports7010019_

Round 1

Reviewer 1 Report

The objectives of the authors were to compare body composition via DXA with absolute and relative isokinetic strength of the knee flexor and extensor muscles of older women with and without involvement in an exercise regimen.

Strengths of the submitted work is the focus on the continued need to better understand the best approach to characterizing muscle performance in older adults and the effort to highlight the consequences of poor body composition within their sample.

In general, the areas of the paper that require further attention center on the rationale and objectives of the study.  A clear understanding of the authors intent was obfuscated by the potential conflation between physical activity and exercise. While the scope of physical activity is broad, there is a known methodology to document it and estimate elements of frequency or 'intensity' (via METS, etc.). Whereas, exercise is a purposeful activity subject to specific adaptions that may far exceed what is observed from one's customary 'physical activity'. Critically, how the authors interpret the published literature hinges on their view of these terms.

Notably, the published literature has no shortage on papers that detail the response of older adults (with and without sarcopenia). Therefore, it is incumbent on the authors to adequately convey the compelling rationale for their work. If the work is indeed a 'secondary study' (which is not unusual and would be fine), then they need to focus in on what can be gleaned from their data.

In my view, the useful info is not the differences in body composition between the exercise and non-exercise groups. In contrast, I think that it is worth highlighting the meaningful differences in the relative strength between the groups. After all, the accepted diagnostic strength criterion for all of the major sarcopenia consensus groups includes measures of absolute values of strength - not relative strength.  (Excluding the FNIH cut off values for grip strength adjusted for BMI which are not used by the major U.S. or European study groups.) However, the clinical usefulness of the study findings would hold more weight if additional functional performance or health-related quality of life data were presented and subject to analysis.

Finally, the discussion loses some ground when the muscle quality concept is featured. The data collected does not include tissue composition measures, and the authors do not really examine the non-significant findings that they observed in the strength data adjusted for lean mass. As mentioned in the detailed comments (see the appended pdf document), they may want to consider published accounts of the changes in muscle tissue composition that occur in older individuals independent of of body fatness. This added information would help to explain their observations, and add much needed context to the limits of their study design. (Especially since the strength training component in the exercising group appears to be somewhat under-dosed, with no clear progression path outlined; which would, perhaps, minimize the differences in muscle tissue composition between the groups).

Please see my pdf file notation on the submitted paper for the line-by-line comments. Thank you for the opportunity to review this work.

Author Response

Manuscript ID: sports-407806

Title: Relative strength, but not absolute muscle strength, is higher in exercising compared to non- exercising older women (former title: Relative strength, but not absolute muscle strength, is higher in physically active compared to inactive older women)

20-Dec-2018

Prof. Dr. Eling Douwe de Bruin

Editor-in-Chief

Sports

Dear Editor,

            We would like to thank the reviewers for their thorough review and insightful feedback; we have made all necessary revisions (highlighted in red) and answered the reviewers’ questions point by point. The manuscript has been improved substantially and we hope it is now suitable for publication in Sports.

# REVIEWER 1

Comments and Suggestions for Authors

The objectives of the authors were to compare body composition via DXA with absolute and relative isokinetic strength of the knee flexor and extensor muscles of older women with and without involvement in an exercise regimen.

Strengths of the submitted work is the focus on the continued need to better understand the best approach to characterizing muscle performance in older adults and the effort to highlight the consequences of poor body composition within their sample.

In general, the areas of the paper that require further attention center on the rationale and objectives of the study.  A clear understanding of the authors intent was obfuscated by the potential conflation between physical activity and exercise. While the scope of physical activity is broad, there is a known methodology to document it and estimate elements of frequency or 'intensity' (via METS, etc.). Whereas, exercise is a purposeful activity subject to specific adaptions that may far exceed what is observed from one's customary 'physical activity'. Critically, how the authors interpret the published literature hinges on their view of these terms.

Answer: Introduction section has been rewritten to clarify the rationale and the aim of the study. The conflation between physical activity and exercise has been corrected. We are aiming to study the effects of an exercise program. Thank you for calling our attention to the conflation between physical activity and exercise. Please let us know if this explanation does not resolve your concerns in this matter.

Notably, the published literature has no shortage on papers that detail the response of older adults (with and without sarcopenia). Therefore, it is incumbent on the authors to adequately convey the compelling rationale for their work. If the work is indeed a 'secondary study' (which is not unusual and would be fine), then they need to focus in on what can be gleaned from their data.

Answer: In fact, the published literature has no shortage on papers that detail the response of older adults; however, we are looking for the exercise training effects on relative muscular strength of older adults, which is an importance measure of the ability to perform functional tasks. Because we rewrote the introduction section, we expect the idea is clearer. Thank you about your constructive comment. Please let us know if this explanation does not resolve your concerns in this matter.

In my view, the useful info is not the differences in body composition between the exercise and non-exercise groups. In contrast, I think that it is worth highlighting the meaningful differences in the relative strength between the groups. After all, the accepted diagnostic strength criterion for all of the major sarcopenia consensus groups includes measures of absolute values of strength - not relative strength.  (Excluding the FNIH cut off values for grip strength adjusted for BMI which are not used by the major U.S. or European study groups.) However, the clinical usefulness of the study findings would hold more weight if additional functional performance or health-related quality of life data were presented and subject to analysis.

Answer: We have rewritten the introduction section for clarification purposes and to meet the reviewer’s expectation. Please let us know if this explanation does not resolve your doubts in this matter.

Finally, the discussion loses some ground when the muscle quality concept is featured. The data collected does not include tissue composition measures, and the authors do not really examine the non-significant findings that they observed in the strength data adjusted for lean mass. As mentioned in the detailed comments (see the appended pdf document), they may want to consider published accounts of the changes in muscle tissue composition that occur in older individuals independent of of body fatness. This added information would help to explain their observations, and add much needed context to the limits of their study design. (Especially since the strength training component in the exercising group appears to be somewhat under-dosed, with no clear progression path outlined; which would, perhaps, minimize the differences in muscle tissue composition between the groups).

Answer: We have rewritten the discussion section in relation to muscle quality, peak torque relative to lean mass and fat infiltration in skeletal mass with aging as requested by you. Thank you about your constructive comment.

Please see my pdf file notation on the submitted paper for the line-by-line comments. Thank you for the opportunity to review this work.

Answer: Below we answer the comments of the reviewer made in the pdf file

Line 4 – The title has been changed as suggested by the reviewer suggestion.

Line 18 – The sample size has been included in abstract as requested by the reviewer.

Line 28 – As Dale et al (2011) recommendation, elderly term has been changed by older persons. Thank you about paying our attention about this important issue.

Line 31 – The phrase has been excluded.

Line 33 – The term “elderly” has been replaced by older adults.

Line 34 - e.g., has been included. 

Line 54 – The paragraph has been rewritten and the study from 2006 has been replaced. The mistake between physical activity and exercise has been corrected and the much attention has been given to the central point of the study witch is relative vs absolute strength assessment. Thank you for your insightful and constructive comment.

Line 58 – The sentence has been rewritten. Thank you for your comment.

Line 60 – The reference has been changed as requested by the reviewer.

Line 66 – The mistake has been corrected.

Line 80 – We totally agree with you. We deleted this information in order to clarify.

Line 124 – We included Capranica’s reference as suggested by the reviewer.

Line 129 – We have changed Nm.kg-1 to Nm•kg-1 in order to clarify.

Line 131 – The subjects was engaged in the program for less than 5 years and more than 6 months. We have included this information in order to clarify.

Line 153 – Volunteers. The word has been replaced in order to clarify.

Line 138 – Unfortunately, we do not have this information. This was added as a study limitation.

Line 140 – Each section was supervised by an experienced coach. This information has been included in order to clarify and meet with the reviewer’s expectation.

Line 154 –  Power analysis is directly related to tests of hypotheses. While conducting tests of hypotheses, the researcher can commit Type II error (probability to falsely infer the absence of something that is present). Statistical power mainly deals with Type II errors. This analysis has been conducted after the data collection, in order to verify if the magnitude of the power level. This explanation has been included in statistical analyses section. Effect size was not calculated.  Please let us know if this explanation does not resolve your doubts in this matter.

Line 155 – All statistical analysis was carried out for two-tailed significance at the 5 % level using Statistica software (Statsoft, Inc., version 6.0 for Windows). We have included this information in order to clarify and meet with the reviewer’s expectation.

Line 164 – Power analysis is directly related to tests of hypotheses, and it deals with Type II error. This information has been included as a table note as suggested by the reviewer. 

Line 236 – The sentence has been rewritten and Miljkovic's observation has been included. Thank you about your constructive suggestion.

Line 243 – The paragraph has been moved to Limitations section. The reviewer suggestion about isokinetic limitation and body composition analysis has been included. Thank you about your suggestion.

Line 255 – The conclusion has been rewritten, after the study premise and purpose are revised.

Reviewer 2 Report

Muscular strength is very important in all age groups, but especially in the elderly population. The present study compared elderly persons who are physically active with an inactive group.

There are some major limitations with the study

-The design is cross-sectional, which is also acknowledged by the authors.

- The total amount of daily physical activity is unknown; no objective measures are included.

However, there are some major strengths also;

-Relative strength is rarely measured and is of great importance.

Suggestions;

The introduction focuses a lot on body composition, but the focus of the paper is muscular strength. It is even more difficult to look at the association between physical activity and body composition. It is likely that people who are obese do less physical exercise and the cross-sectional design makes the causal relationship impossible to analyse.

My suggestion is to focus more on the muscular strength in the introduction. It is very important in the elderly population and most likely associated with overall function and the risk for falls, etc.

Even if it is likely that the exercise improved body composition and muscular strengths the cross-sectional design makes this conclusion impossible.

Author Response

Manuscript ID: sports-407806

Title: Relative strength, but not absolute muscle strength, is higher in exercising compared to non- exercising older women (former title: Relative strength, but not absolute muscle strength, is higher in physically active compared to inactive older women)

20-Dec-2018

Prof. Dr. Eling Douwe de Bruin

Editor-in-Chief

Sports

Dear Editor,

            We would like to thank the reviewers for their thorough review and insightful feedback; we have made all necessary revisions (highlighted in red) and answered the reviewers’ questions point by point. The manuscript has been improved substantially and we hope it is now suitable for publication in Sports.

# REVIEWER 2

Comments and Suggestions for Authors

Muscular strength is very important in all age groups, but especially in the elderly population. The present study compared elderly persons who are physically active with an inactive group.

 There are some major limitations with the study

-The design is cross-sectional, which is also acknowledged by the authors.

Answer: in fact, this is a study limitation. It was discussed in the Study limitation and strengths section.

 - The total amount of daily physical activity is unknown; no objective measures are included.

Answer: In fact, there were a mistake between terms exercise and physical activity as it was pointed by reviewer #1. This mistake has been corrected and the aim of the study was to compare exercising and non-exercising older women. The exercising characteristics were described in the “Training program” section. The total amount of daily exercise was 1 hour per day. Please let us know if this explanation does not resolve your concerns in this matter.

 However, there are some major strengths also;

-Relative strength is rarely measured and is of great importance.

 Answer: Thank you very much for your constructive comment.

Suggestions;

 The introduction focuses a lot on body composition, but the focus of the paper is muscular strength. It is even more difficult to look at the association between physical activity and body composition. It is likely that people who are obese do less physical exercise and the cross-sectional design makes the causal relationship impossible to analyse.

Answer: Thank you very much about the suggestion. The introduction section has been rewritten in order to clarify. Please let us know if this explanation does not resolve your concerns in this matter.

My suggestion is to focus more on the muscular strength in the introduction. It is very important in the elderly population and most likely associated with overall function and the risk for falls, etc.

Answer: We agree with the reviewer about the importance of muscular strength. The introduction section has been rewritten. Please let us know if this explanation does not resolve your concerns in this matter.

Even if it is likely that the exercise improved body composition and muscular strengths the cross-sectional design makes this conclusion impossible.

Answer: In fact, the authors agree with the reviewer. In order to be careful with the conclusions in a cross-section design, the authors avoid affirming that the exercise improved body composition and muscular strengths, instead of this; the discussion was based only in inter-groups differences.   

Reviewer 3 Report

A very interesting research problem undertaken by the authors of the article. However, if,  "The AG was composed of women who were physical active for at least 6 months prior to study commencement. They held training sessions 3 times a week, 1 h per session.” (ver. 132-133) then, the comparisons presented in the section on the analysis of results take into account the results of the AG group before or after the training. The article is not clearly specified. The thought that it is worth completing the analysis by comparing the results achieved by the AG group before and after training, and compare these results with the IG group. Such a comparison would give a picture of the impact of the implemented training on changes in the composition of the body, strength and other measured indicators.
In addition, the authors do not describe how they calculated peak torque relative to fat-free mass, and relative isokinetic peak torque. Perhaps this is obvious, but for scientific correctness such information should in my opinion be included in the content of the article.

Author Response

Manuscript ID: sports-407806

Title: Relative strength, but not absolute muscle strength, is higher in exercising compared to non- exercising older women (former title: Relative strength, but not absolute muscle strength, is higher in physically active compared to inactive older women)

20-Dec-2018

Prof. Dr. Eling Douwe de Bruin

Editor-in-Chief

Sports

Dear Editor,

            We would like to thank the reviewers for their thorough review and insightful feedback; we have made all necessary revisions (highlighted in red) and answered the reviewers’ questions point by point. The manuscript has been improved substantially and we hope it is now suitable for publication in Sports.

# REVIEWER 3

Comments and Suggestions for Authors

A very interesting research problem undertaken by the authors of the article. However, if,  "The AG was composed of women who were physical active for at least 6 months prior to study commencement. They held training sessions 3 times a week, 1 h per session.” (ver. 132-133) then, the comparisons presented in the section on the analysis of results take into account the results of the AG group before or after the training. The article is not clearly specified. The thought that it is worth completing the analysis by comparing the results achieved by the AG group before and after training, and compare these results with the IG group. Such a comparison would give a picture of the impact of the implemented training on changes in the composition of the body, strength and other measured indicators.

In addition, the authors do not describe how they calculated peak torque relative to fat-free mass, and relative isokinetic peak torque. Perhaps this is obvious, but for scientific correctness such information should in my opinion be included in the content of the article.

Answer: The authors agree with the reviewer. If the paper was a clinical trial, differences between pre and post training could be analyzed. However, the paper was a cross-sectional study, so it is possible to conclude only about difference between groups. The cross sectional study design was included as a study limitation. A clearer information about the experimental design has been included in order to clarify and meet with the reviewer’s expectations

Peak torque relative to total body mass and relative to fat-free mass were calculated dividing peak torque (Nm) values to total body mass (Kg) and to fat-free mass (KgFFM), respectively. This information has been included in order to clarify and meet with the reviewer’s expectation.